# Thermo-Mechanical Approach to Material Extrusion Process During Fused Filament Fabrication of Polymeric Samples

**DOI:** 10.3390/ma18194537

**Published:** 2025-09-29

**Authors:** Mahmoud M. Farh, Viktor Gribniak

**Affiliations:** 1Department of Steel and Composite Structures, Vilnius Gediminas Technical University (VILNIUS TECH), Saulėtekio Av. 11, 10223 Vilnius, Lithuania; mahmoud-mohammed.farh@vilniustech.lt; 2Laboratory of Innovative Building Structures, Vilnius Gediminas Technical University (VILNIUS TECH), Saulėtekio Av. 11, 10223 Vilnius, Lithuania

**Keywords:** finite element modeling, fused filament fabrication (FFF), polylactic acid (PLA), prototyping, residual stresses, thermo-mechanical analysis, warpage

## Abstract

While material extrusion via fused filament fabrication (FFF) offers design flexibility and rapid prototyping, its practical use in engineering is limited by mechanical challenges, including residual stresses, geometric distortions, and potential interlayer debonding. These issues arise from the dynamic thermal profiles during FFF, including temperature gradients, non-uniform hardening, and rapid thermal cycling, which lead to uneven internal stress development depending on fabrication parameters and object topology. These problems can compromise the structural integrity and mechanical properties of FFF parts, especially when the load-bearing capacity and geometric accuracy are critical. This study focuses on polylactic acid (PLA) due to its widespread application in engineering. It introduces a computational framework for coupled thermo-mechanical simulations of the FFF process using ABAQUS (Version 2020) finite element software. A key innovation is an automated subroutine that converts G-code into a time-resolved event series for finite element activation. The simulation framework explicitly models the sequential stages of printing, cooling, and detachment, enabling prediction of adhesive loss and post-process warpage. A transient thermal model evaluates the temperature distribution during FFF, providing boundary conditions for a mechanical simulation that predicts residual stresses and warping. Uniquely, the proposed model incorporates the detachment stage, enabling a more realistic and experimentally validated prediction of warpage and residual stress release in FFF-fabricated components. Although the average deviation between predicted and measured displacements is about 10.6%, the simulation adequately reflects the spatial distribution and magnitude of warpage, confirming its practical usefulness for process optimization and design validation.

## 1. Introduction

Additive manufacturing (AM) has transformed modern engineering production by enabling the fabrication of complex geometries with high customization potential [1]. Among various AM techniques, fused filament fabrication (FFF) is notable for its accessibility and versatility in processing thermoplastic polymers [2]. This technology is also referred to as fused deposition modeling (FDM) and material extrusion (MEX) with filaments. Although “FDM” remains predominant in the scientific literature, it is a trademarked term that does not fully capture the technical specificity of the fabrication process. FFF explicitly denotes the extrusion-based 3D printing process using thermoplastic filaments, whereas FDM is sometimes used generically and may introduce ambiguity. According to ISO/ASTM 52900 [3], “material extrusion (MEX)” is the standardized process category, while FFF provides a precise designation for the process addressed in this study. Therefore, the terminology “FFF” is retained throughout this manuscript.

Despite its widespread use, FFF faces notable limitations in structural applications due to mechanical issues, including residual stresses, geometric inaccuracies, and poor interlayer adhesion. These problems primarily arise from the complex thermal cycles during fabrication, where the polymer transitions from a semi-molten to a solid state, resulting in internal stresses and distortions [4]. Addressing these effects necessitates simulation frameworks capable of modeling the sequential deposition and thermal evolution of the material, as shown in recent finite element studies [5].

The mechanical performance of FFF components is susceptible to thermal gradients and cooling rates, which are influenced by process parameters such as nozzle temperature, printing speed, and layer thickness [6,7,8,9]. Finite element (FE) simulations serve as a powerful predictive tool for analyzing stress development and deformation patterns. However, experimental optimization remains resource-intensive, underscoring the need for robust models that can simulate the FFF process with high fidelity [10,11]. A key advancement in this domain is the implementation of element activation strategies within the FE modeling framework, enabling the progressive construction of the FFF part in a virtual environment [12]. This approach captures the sequential nature of material deposition and the evolving thermal and mechanical states of the structure [13]. Early studies by Zhang and Chou [14,15] demonstrated the effectiveness of this method in analyzing the formation of residual stress in acrylonitrile butadiene styrene (ABS) components. Cattenone et al. [16] emphasized the importance of accurate constitutive modeling for predicting distortions.

Building on these foundations, recent research has explored more advanced activation schemes to enhance the fidelity of numerical simulations in AM. For instance, Syrlybayev et al. [5] and Khanafer et al. [17] applied the element birth and death method proposed by Zhang and Chou [14] to estimate the warpage potential of FFF samples and investigate the relationship between the FFF process settings and their impact on the fabrication quality of polymeric components. Courter et al. [18] and Barocio et al. [19] integrated time-dependent filament paths and thermal histories into their simulations, facilitating a realistic representation of the sequential material deposition process. These studies highlight the growing importance of G-code-driven simulations, where the toolpath data directly informs the activation sequence of finite elements. Typically, G-code determines a routine that transforms a numerical (computer-aided design, CAD) model into a physical object via FFF.

Further advancements have been achieved by integrating finite element activation with process-aware control strategies. Matúš et al. [20] proposed a methodology that utilizes FE modeling-based stress analysis to inform slicing and deposition strategies, dynamically adjusting infill density and the orientation of pathways based on local stress distributions. This approach not only enhances structural performance but also improves material efficiency and FFF productivity. Furthermore, the element activation technique has evolved from basic geometric triggers (e.g., element centroid within a deposition path) to more sophisticated criteria involving thermal thresholds or mechanical state variables. This progression enables better modeling of phenomena such as layer bonding, cooling-induced shrinkage, and interlayer stress accumulation, all of which are crucial for ensuring the quality of the fabricated part.

Table 1 provides a comparative overview of recent FE modeling studies [5,21,22,23,24,25,26] that address thermomechanical behavior and warpage in extrusion-based additive manufacturing. The table summarizes the simulation software used, the scope of each study, whether progressive element activation (i.e., sequential simulation of material deposition noted as “Activation”) was implemented, and whether the detachment of the printed part from the build platform (indicated as “Detachment”) was explicitly modeled, and it provides the type of outputs and experimental validation reported. Notably, while progressive activation is common, most studies simulate only the printing and cooling phases, assuming perfect adhesion to the build platform throughout. Explicit modeling of the detachment stage—critical for realistic prediction of warpage and residual stress release—remains rare. The present study uniquely incorporates all three stages (printing, cooling, detachment) and provides experimental validation for PLA components.

Although recent advances in finite element modeling have improved the simulation of extrusion-based additive manufacturing, most prior studies have been limited to the printing and cooling phases, without explicit consideration of the detachment stage (Table 1). This omission limits the ability to predict warpage and the release of residual stress realistically. This study introduces a novel computational framework that significantly advances the concept discussed above. The proposed methodology uniquely features a staged simulation approach that explicitly models the sequential printing, cooling, and detachment phases, addressing a critical gap in the literature. Unlike previous methods that rely on simplified or manually defined activation patterns, the proposed methodology features an automated subroutine that parses G-code instructions and converts them into a time-resolved event series for element activation within AM Modeler using FE software ABAQUS (DASSAULT SYSTEMS, Version 2020, Providence, RI, USA). While similar G-code-based activation strategies have recently emerged in the literature, the explicit simulation of the detachment stage distinguishes this work. It enables a more comprehensive and experimentally validated prediction of deformation mechanisms in PLA components fabricated via FFF. The modeling framework couples a transient thermal analysis with a mechanical simulation to predict residual stresses and warpage in polylactic acid (PLA) components fabricated via FFF. Physical tests verify the model’s predictive capability, highlighting its potential for optimizing FFF parameters and improving part quality. By integrating G-code-based element activation with thermo-mechanical modeling, this study provides a comprehensive and scalable tool for controlling FFF processes.

## 2. Research Context

The recent drone prototyping project [27] motivated the development of the present study. That project developed an FFF model of the drone wing for manufacturing with a desktop 3D printer. The model had to maintain the drone’s weight, which was designed from expanded polyethylene. This constraint imposed strict weight limitations. The fabrication space limitations of the Prusa i3 MK3 3D printer (PRUSA RESEARCH, Prague, Czech Republic) necessitated fragmentation during the slicing stage of the FFF process. Figure 1 illustrates the slicing strategy applied to the drone prototype, while Figure 2 highlights the fabrication flaws encountered during the printing process. The weight limitations resulted in a reduction of the wall thickness in the wing fragments and the rejection of the support surfaces in the sliced model (Figure 2a). However, this geometric reduction (i.e., insufficient bonding area) resulted in inadequate adhesion to the printing bed, ultimately causing fabrication failure, as shown in Figure 2b. Increasing the bonding area addressed this issue [27]. Still, the debonding of the polymeric parts (e.g., Figure 2b) prompted the development of a warpage prediction model tailored to simulate differential layer deformation in 3D-printed polymeric components. This explicit consideration of the detachment phase distinguishes the present work from previous studies and provides a more realistic assessment of failure risks in FFF-produced components.

The present study addresses this problem by introducing a staged thermo-mechanical simulation framework that explicitly incorporates the printing stage, the cooling stage, and, critically, the detachment stage. By modeling the loss of adhesive contact between the build platform and the fabricated sample during detachment, the framework enables realistic prediction of warpage and deformation mechanisms that occur during and after this phase in FFF-fabricated components.

This model simulates the differential deformation of layers within 3D-printed polymeric parts, using the element birth and death concept [14]. A transient thermal model determines the temperature distribution throughout the printing process, providing a boundary condition for the subsequent mechanical simulation. The development of an automatic G-Code generation procedure into an Event Series, which describes the nozzle’s position as a function of time and activates the corresponding elements, significantly contributes to this study’s impact on engineering practice. Unlike earlier approaches that relied on simplified geometries and manually defined activation sequences [5,14,18], the present study introduces a scalable simulation workflow that includes a detachment simulation stage. This advancement facilitates the direct integration of slicing data into element activation, thereby enhancing the adequacy and computational efficiency of thermo-mechanical modeling in FFF.

## 3. Materials and Methods

This research builds upon the findings of a prior program [27] that investigated additive drone manufacturing using a desktop 3D printer and commercially available polymeric materials. The comparative analysis presented by Šostakaitė et al. [27] outlines key material properties, including tensile strength, modulus of elasticity, and thermal characteristics. These insights, together with PLA’s compatibility with desktop FFF printers, justify its selection for this study, given its non-toxicity, ease of processing, and affordability. Additionally, recent studies have reinforced the appropriateness of PLA-based materials for drones, highlighting their favorable strength-to-weight ratio, ease of printing, and suitability for complex geometries [28,29].

### 3.1. Specimen Design and Fabrication Parameters

The experimental phase employed standard tensile specimens conforming to the ASTM D638-14 geometry [30], which is widely used for evaluating the mechanical properties of polymeric materials. The test specimens were produced using a Prusa i3 MK3 printer and sliced with PRUSASLICER 2.3.3 (PRUSA RESEARCH, Prague, Czech Republic), which offers the resolution and control required for research-grade additive manufacturing. Figure 3 illustrates the research workflow.

The slicing parameters were selected to ensure complete material deposition and minimize interlayer defects, based on prior studies that demonstrated their effectiveness in producing mechanically robust PLA components [4,5,6,8,27]. For instance, Šenkeřík et al. [4] analyzed the influence of extrusion parameters on PLA filament quality, while Mosleh et al. [6] simulated thermal profiles under similar conditions. The experimental phase of this study was designed in direct response to the fabrication challenges identified in the preceding research context (Section 2), particularly those related to inadequate adhesion and warpage during FFF. Building on these findings, the present work applies a staged thermo-mechanical simulation framework—previously detailed—to systematically investigate the thermal and mechanical evolution of printed parts under constrained geometries and weight-sensitive applications.

The literature analysis [27] substantiated the material choice for this investigation. The PLA filament had a nominal diameter of 1.75 ± 0.02 mm, as specified by the manufacturer. According to the manufacturer’s technical datasheet [31], the allowable fabrication parameters include a nozzle temperature of 210 ± 10 °C, a heated bed temperature ranging from 40 °C to 60 °C, and a print speed of up to 200 mm/s. A cooling fan speed of 100% is advised to ensure optimal print quality. The previous tests [27,32,33] defined the specific printing settings to ensure the sample’s warpage during physical testing.

To induce warpage, the specimens were printed on a replacement spring steel sheet coated with smooth, double-sided PEI (polyethylenimine), without additional adhesion measures. The physical experiment was conducted in an open-chamber environment to simulate the printing conditions shown in Figure 2b. After fabrication and cooling under laboratory conditions (at 20 °C) until the printing bed temperature reached 25 °C, the test samples were removed by bending (flexing) the spring steel sheet to mechanically detach the specimen, where the specimen edge experienced debonding due to the cooling-induced release.

Thus, in the first fabrication trial, the layout of the dumbbell-shaped test sample included two solid shells along the printing perimeter; the inner part of the specimen was printed in 11 layers, yielding a total specimen thickness of 3.2 mm. The first layer height was 0.2 mm, with subsequent layers at 0.3 mm each, and a 100% infill oriented at a ±45° angle, alternating the infill angle between layers. Printing was conducted using a 0.4 mm nozzle at 210 °C and a speed of 30 mm/s; the printing bed temperature was maintained at 60 °C. Warpage was observed at the microscale in the 3D printed part.

To observe warpage at the macroscale, the second fabrication stage utilized the same fabrication parameters as the first stage, but with a reduced thickness of 0.4 mm, printed in two layers of 0.2 mm each. This modified geometry was selected to facilitate observation of warpage during physical testing.

### 3.2. G-Code Conversion and Element Activation

A central innovation of this study is the automated conversion of G-code into a time-resolved event series compatible with FE simulations in ABAQUS (SIMULA, DASSAULT SYSTEMS, Version 2020, Providence, RI, USA). G-code, which encodes the printer’s toolpath and associated process parameters, is parsed using a custom PYTHON script. This script extracts the time-dependent filament centerlines and deposition sequences from the G-code file—generated by slicing software—and converts them into an event series that defines the activation intervals of finite elements within the simulation domain. The resulting event series specifies the nozzle’s position in the *x*, *y*, and *z* axes over time, as well as the extrusion status, as shown in Table 2. The *x*, *y*, and *z* columns specify the spatial coordinates of the extruded filament centerline at time *t*. The final column indicates the deposition status, where 1 denotes active extrusion, and 0 represents non-depositing movements. Since the deposition path in the G-code is not always continuous, the nozzle may perform non-depositing movements during travel. The filament cross-section was approximated as a rectangle, with dimensions matching the major and minor axes of the elliptical profile shown in Figure 4.

The element activation strategy employs a progressive element birth approach, wherein elements are activated when their centroids intersect the defined deposition path. This method, initially introduced by Zhang and Chou [14], has been widely adopted in FFF simulations due to its capability to replicate the sequential nature of material deposition. In this study, the filament’s elliptical cross-section was approximated as a rectangle, and a binary state variable was assigned to each element, indicating activation (1) or inactivity (0) at a specific time.

The generated G-code includes the necessary instructions for the 3D printer, such as printing speed, extrusion temperature, and print path. As previously described, the custom PYTHON script transforms this G-code into an event series for FE analysis. Figure 4 illustrates the element activation process, which is based on the time-dependent filament coordinates provided by the event series described in Table 2. As shown in Figure 4, the filament’s elliptical cross-section is approximated as a rectangle. Within the simulation, an element is activated and included in the analysis if its center lies within these rectangular boundaries.

This approach is consistent with the methodology used by Syrlybayev et al. [5], who applied a similar element activation scheme to simulate warpage in FFF parts. Their work demonstrated that accurate modeling of the deposition sequence significantly improves the prediction of thermal gradients and mechanical distortions. The present study extends this concept by integrating the event series directly into the FE simulation framework, enabling a seamless transition from slicing to simulation.

Automating the G-code conversion significantly reduces manual preprocessing efforts, enhancing the reproducibility and scalability of simulation workflows. This capability is particularly valuable for parametric studies and optimization tasks, where multiple simulations must be conducted under varying process conditions.

### 3.3. Thermal Simulation

In the FFF process, the filament is extruded at a temperature of *T_f_* and deposited onto a heated build platform maintained at a temperature of *T_b_*. The classical heat conduction equation governs the transient thermal behavior of the printed part [34]:(1)$$\rho c\frac{𝜕T}{𝜕t}=\nabla \;\cdot \;\left(k\nabla T\right)+q,$$
where *ρ* is the material density; *c* is the specific heat capacity; $\frac{𝜕T}{𝜕t}$ is the time rate of change of temperature; *T* is the temperature; ∇ is the divergence operator; *k* is the thermal conductivity; *q* is the internal heat source.

The initial temperature distribution is defined as follows [35]:(2)$$\begin{array}{cc} T\left(x,0\right)=T_{f}, & x\;\in \;\Omega_{f} \end{array},$$(3)$$\begin{array}{cc} T\left(x,\;0\right)=T_{b}, & x\;\in \;\Omega_{b} \end{array},$$
where the vector **x** = {*x*, *y*, *z*} determines the position; Ω*_f_* is the boundary of the extruded and deposited material; Ω*_b_* is the surface of the print bed; *T_f_* and *T_b_* are the extrusion and print bed temperatures. The Neumann boundary conditions, accounting for convective and radiative heat exchange, are expressed as follows [35]:(4)$$\begin{array}{cc} k_{f}\frac{𝜕T}{𝜕n}+q_{c}+q_{r}=0, & x\in S\left(t\right) \end{array},$$
where *S*(*t*) is the external surface of the body (changing during the element activation); **n** is the vector normal to the surface of the body; *q_c_* and *q_r_* represent the convective and radiative heat fluxes, which are defined as follows:(5)$$q_{c}=\;h\left(T-T_{s}\right);$$(6)$$q_{r}=K_{b}\left(T^{4}-T_{s}^{4}\right)e,$$
where *h* is the heat transfer coefficient; *K_b_* represents the Stefan–Boltzmann constant; *T_s_* is the surrounding temperature; *e* is the emissivity. Table 3 summarizes the temperature-dependent physical properties of PLA used in the thermal simulation, as collected from references [36,37,38]. In this table, *T* is the material temperature; *E* is the deformation modulus; *ν* is Poisson’s ratio; *σ_y_* is the yield strength; *α* is the thermal expansion coefficient; *k* is the thermal conductivity; *c* is the specific heat capacity; *ρ* is the density.

The thermal simulation used eight-node linear heat transfer brick elements (DC3D8) with ABAQUS (SIMULA, DASSAULT SYSTEMS, Version 2020, Providence, RI, USA). Figure 5 illustrates the thermal problem considered in this work. This formulation requires specifying the temperature-dependent thermal properties of the PLA filament, as well as the filament and printing bed temperatures (*T_f_* and *T_b_*), the emissivity (surface radiation) factor (*e*), the heat transfer coefficients of the filament and build platform (*h_f_* and *h_b_*), and the surrounding and room temperatures (*T_s_* and *T_a_*). The simulation comprised three distinct phases: printing, cooling, and detachment. During the printing stage, newly activated elements were initialized at the extrusion temperature of 210 °C, while the build platform was maintained at 60 °C. These boundary conditions were applied exclusively during the printing phase and removed during cooling and detachment to replicate the physical release of the printed part.

Heat transfer was modeled using the temperature-dependent thermal properties of PLA, as reported in previous studies [5,17]. The film coefficients were set at 72 W/m^2^·°C during printing and 67 W/m^2^·°C during cooling, while the emissivity values were 0.92 and 0.0, respectively. These values were selected based on experimental data and validated modeling practices in the literature. For example, Khanafer et al. [17] used similar thermal parameters to simulate heat flow in FFF processes, emphasizing the importance of accurate boundary conditions in predicting thermal gradients.

The thermal model also considered conduction between the filament and the build platform, as well as between adjacent filament layers. Convection to the surrounding environment was included to simulate natural cooling. The detachment stage assumed zero heat exchange, representing the removal of the part from the printer and its exposure to ambient conditions. This staged thermal modeling approach is consistent with the methodology proposed by Cattenone et al. [16], which enables the comprehensive simulation of residual stress evolution and post-printing deformation. The present study builds on this by explicitly modeling the detachment phase, which is critical for understanding warpage behavior in unsupported regions of the part.

### 3.4. Mechanical Simulation

In Abaqus, engineering quantities are processed in Voigt notation rather than tensor form, with internal tensor-to-matrix transformation (Voigt mapping) performed automatically by the software [39]. Thus, it transforms the assumed physical properties (Table 3) into the second-order and fourth-order tensor mappings internally, building the stiffness matrix *S*(*T*). Cattenone et al. [16] incorporated the computed time-dependent temperature distribution as externally prescribed, solution-independent conditions into a thermo-elastic–plastic constitutive model to evaluate the total strain, which is expressed as:(7)$$\varepsilon =\varepsilon_{e}+\varepsilon_{pl}+\varepsilon_{t},$$
where *ε* is the total strain, and *ε_e_*, *ε_pl_*, and *ε_t_* are the elastic, plastic, and thermal strain components, with individual components defined as follows:(8)$$\varepsilon_{e}=S\left(T\right)\cdot \sigma ,$$(9)$$\varepsilon_{pl}=\lambda \sigma_{dev},$$(10)$$\varepsilon_{t}=\alpha \left(T\right)\nabla T,$$
where *σ* is the stress in Voigt notation; *S*(*T*) defines temperature-dependent compliance (the inverse of stiffness matrix); *σ_dev_* represents the deviatoric part of the stress tensor; *α*(*T*) is the temperature-dependent second-order coefficient of thermal expansion tensor; *λ* is the plastic flow factor, determined according to the following criterion [40]:(11)$$\lambda \left\{\begin{array}{c} \begin{array}{cc} =0, & \sigma_{vm\;}<\;\sigma_{y}\left(T\right); \end{array}\\ \begin{array}{cc} >0, & \;\sigma_{vm\;}\geq \;\sigma_{y}\left(T\right), \end{array} \end{array}\right.$$
where *σ_y_* is the yield stress, and *σ_vm_* is the effective Von Mises stress that can be described as follows:(12)$$\;\sigma_{vm\;}=\sqrt{\frac{2}{3}{\left(\sigma_{dev}\right)}^{T}\sigma_{dev.}}$$
where the operator (□)^T^ indicates transposing the deviatoric part of the stress tensor. Thus, Equation (11) describes two cases: (1) The elastic behavior of the material when the equivalent stress is below the yield stress; this means no plastic deformation occurs, and therefore, the plastic flow multiplier (*λ*) is zero, keeping the stress state within the yield surface boundary. (2) The material undergoes plastic deformation when the equivalent stress reaches or exceeds the yield stress. In this case, the stress state is on the yield surface.

The mechanical simulation employed 20-node quadratic brick elements (C3D20R), utilizing the temperature distribution from the thermal model as a solution-independent boundary condition. During printing and cooling, the bottom surface of the specimen (Figure 3) was constrained in all translational degrees of freedom (*u_x_* = *u_y_* = *u_z_* = 0) to simulate adhesion to the build platform. Figure 6 shows the boundary constraints. These constraints were removed during the detachment phase to simulate the release of residual stresses and the onset of warpage. The mechanical simulation predicted residual stress accumulation and geometric deformation, with maximum displacement localized at the specimen edges due to non-uniform cooling. This behavior aligns with the findings of Zhang and Chou [15], who noted similar deformation patterns in ABS components. The current study confirms that PLA exhibits similar behavior, with stress accumulation during cooling and relaxation upon detachment.

A convergence study was conducted to determine the optimal mesh density and time step. The convergence study revealed that a mesh resolution of one element per filament cross-section yields adequate accuracy, with minimal variation in predicted warpage. This finding aligns with the recommendations of Barocio et al. [19], who emphasized the importance of mesh refinement in capturing localized thermal and mechanical gradients. The mechanical model also assesses the effect of the coefficient of thermal expansion on the driving deformation of the FE model. As noted by Espinach et al. [37], PLA exhibits significant shrinkage upon cooling, which must be adequately represented in the model to ensure reliable warpage prediction. The current study incorporates temperature-dependent material properties to enhance the simulation’s fidelity.

### 3.5. Experimental Validation

The following simulations focus on the detachment stage, which, as demonstrated in the preceding sections (Figure 2), can significantly influence the deformation and warpage of FFF-produced components under certain fabrication conditions. By explicitly modeling the loss of adhesive contact between the printed part and the build platform, the proposed framework enables a realistic assessment of deformation mechanisms that may arise during and after detachment. This capability is particularly relevant for evaluating the dimensional accuracy and structural integrity of components produced under challenging adhesion scenarios.

Experimental validation was performed using the 0.4 mm thick specimen fabricated during the second stage, as detailed in Section 3.1. Validation of the numerical model involved measuring vertical displacements at 11 predefined locations along the printed specimen using a Vernier caliper with a resolution of 0.02 mm. Figure 7a identifies the measurement points for warpage assessment. Figure 7b presents the numerical simulation results, while Figure 7c provides a comparative analysis between experimental and simulated deformation profiles. The presented results correspond to an FE model comprising 27,512 elements.

Figure 8 illustrates the influence of mesh density on the computational efficiency of the warpage simulation. The simulations were performed by varying the number of FE in the model (Figure 7b) from 9000 to 548,720, executed on a workstation equipped with 16 GB RAM and an AMD Ryzen 9 5900HX processor at 3.30 GHz. Additional FE simulations were performed to evaluate the sensitivity of warpage to variations in extrusion temperature. Increasing the temperature from 200 °C to 220 °C, as recommended by the manufacturer’s datasheet [31]. Within the manufacturer’s allowable range, the selected temperatures reflect a practical scenario encountered in desktop FFF applications. These simulations yielded a 6.7% increase in maximum displacement, underscoring the notable influence of thermal input on deformation behavior. This finding corroborates the conclusions of Liu et al. [8], who emphasized the critical role of thermal history in determining the final geometry of printed components.

## 4. Results and Discussion

### 4.1. Warpage and Residual Stress Analysis

A key innovation of the present study is the development of a staged thermo-mechanical simulation framework that explicitly models the sequential stages of the fused filament fabrication process: printing, cooling, and detachment. The explicit simulation of the detachment phase distinguishes this framework from previous approaches [5,21,22,23,24,25,26], which typically focus solely on the printing and cooling stages. By incorporating the detachment stage, the model enables a more realistic prediction of warpage and residual stress release, as confirmed by experimental validation. This comprehensive simulation concept, experimentally verified for PLA specimens, provides new insights into the evolution of residual stresses and warpage throughout the entire fabrication process and establishes a robust basis for predictive modeling and process optimization in extrusion-based AM.

Applying this framework, the developed thermo-mechanical modeling framework (Figure 3) enables efficient prediction of warpage behavior in PLA specimens fabricated via the FFF process. The simulation results (Figure 7) indicate that residual stresses were negligible during the printing phase but increased significantly during cooling due to thermal contraction and substrate constraints. Upon detachment, the release of these constraints caused a sudden relaxation of residual stresses, leading to observable warping, predominantly at the specimen edges. This observation underscores the importance of explicitly modeling the detachment stage within the simulation framework, as it is critical for capturing the dominant deformation mechanisms and failure risks associated with the loss of adhesive contact between the printed part and the build platform.

The experimentally measured maximum vertical displacement was 6.82 mm, which aligns closely with the simulation results. The average deviation between the numerical and experimental results was approximately 10.6%, indicating that the model has adequate predictive capability. This level of agreement aligns with findings by Cattenone et al. [16] and Armillotta et al. [41], who reported a dimensional deviation of 12% in similar simulations involving ABS materials. Comparable studies by Syrlybayev et al. [5] and Barocio et al. [19] achieved prediction errors ranging from 8% to 15%, depending on the complexity of the geometry and the thermal boundary conditions applied. In this context, the 10.6% deviation observed in the present study falls within the typical range reported for thermo-mechanical simulations of FFF processes, confirming the model’s capability to capture the dominant deformation mechanisms. Moreover, the spatial distribution of warpage predicted by the simulation closely matches the experimental observations, which is essential for applications where localized distortions may impair structural performance or dimensional fidelity. The agreement between simulated and experimental warpage profiles demonstrates the adequacy of the staged simulation approach—particularly the inclusion of the detachment phase—for capturing the dominant deformation mechanisms in FFF-fabricated components.

The current study extends the understanding of warpage behavior in PLA materials, highlighting the critical role of thermal contraction and constraint release. This insight is particularly relevant for optimizing the FFF process to minimize warpage and improve part quality. The findings also underscore the importance of accurate thermal modeling in predicting residual stresses and deformation patterns. The automated generation of event series from G-code enhances the versatility of simulations. It reduces manual intervention, avoiding the errors characteristic of manually updating the model (e.g., [5,17,18]) and thereby speeding up the simulation process. These findings suggest that future research should focus on refining the thermal model to account for material-specific properties and environmental conditions. Additionally, exploring the effects of different build platform materials and adhesion techniques could provide further insights into minimizing warpage and enhancing the overall quality of 3D-printed parts.

### 4.2. Convergence and Sensitivity Studies

A convergence study was conducted to determine the optimal mesh resolution and time-stepping parameters. The results indicated that using one element per filament cross-section yielded sufficient accuracy, with a variation of less than 2.1% in predicted warpage across mesh densities. Similarly, a time step of 0.01 s during the printing stage and 0.1 s during cooling and detachment was found to strike a balance between computational efficiency and predictive fidelity. These findings align with recommendations by Syrlybayev et al. [5], Barocio et al. [19], and Brenken et al. [21], who emphasized the importance of mesh refinement and time-step control in capturing localized thermal and mechanical gradients. The extrusion temperature also significantly influenced warpage. Increasing the temperature from 200 °C to 220 °C produced a 6.7% increase in maximum displacement, highlighting the sensitivity of warpage to extrusion temperature. This observation supports the conclusions of Liu et al. [8], who highlighted the role of thermal history in determining final part geometry.

The mesh size sensitivity analysis (Figure 8) shows that computational time, for both thermal and mechanical analyses, increases linearly with the number of elements. While refining the mesh significantly improves the accuracy of stress gradient predictions, especially when the mesh resolution closely matches filament dimensions, it substantially increases computational expense. Therefore, to obtain sufficiently accurate results, it is recommended to use one element per filament’s cross-section and a 0.02 s time step (Δ*t*) for efficient computation, which corresponds to 2220 s CPU time and activation temperature of 192 °C, as shown in Figure 8b.

At the same time, developing an appropriate meshing strategy is complex and must be customized to the specific needs of the analysis. While a finer mesh improves the accuracy of stress gradient predictions, it also significantly increases computational costs. This contradiction is apparent in mechanical analyses, where stress concentrations raise the system’s sensitivity to mesh refinement. Thus, the increase in accuracy observed in Figure 8b does not seem to converge as the number of elements grows, which may result from local stress singularities or the continued rise in stress with mesh refinement, ensuring that the stress state remains within or on the yield surface, depending on the magnitude of the equivalent stress. Additionally, uniformly refining the mesh without targeted focus on critical stress regions is inefficient and raises computational time. In some cases, nonlinear material behavior can also influence this trend [42,43]. Therefore, for smaller models—with significant local stress effects and manageable computational costs—a finer mesh is advisable for capturing localized mechanical responses in small-scale models. Conversely, for larger models where local variations are less significant relative to the overall geometry, a coarser mesh is generally more suitable for reducing computational demands while maintaining acceptable accuracy.

The current study enhances the existing body of knowledge by offering detailed convergence and sensitivity analyses for the FFF process. These findings are essential for optimizing process parameters and ensuring the accuracy and reliability of simulation results. The insights gained from this study can be applied to enhance the design and manufacturing of 3D-printed parts, particularly in applications that require high precision and structural integrity. Future research should examine the effects of other process parameters, such as print speed and layer height, on warpage and residual stresses. Furthermore, investigating the interactions between various parameters could yield a more comprehensive understanding of the FFF process and its influence on part quality.

### 4.3. Comparison with Previous Studies, Limitations of the Model, and Further Research

The proposed model builds upon and advances earlier work by Zhang and Chou [14,15], who introduced the element birth and death method for simulating residual stresses in ABS components. Unlike those studies, the proposed staged modeling strategy, which incorporates the sequential simulation of printing, cooling, and detachment, enables a more comprehensive assessment of residual stress evolution and warpage than approaches limited to the printing and cooling phases. This innovation ensures the direct incorporation of slicing parameters and toolpath data into the FE model and simulates the post-printing stage, thereby enhancing the numerical prediction’s fidelity.

In terms of geometric complexity, previous studies, such as those by Syrlybayev et al. [5] and Armillotta et al. [41], primarily addressed simplified geometries with manually defined activation sequences. The current workflow (Figure 3) demonstrates the feasibility of simulating complex deposition paths through a fully automated workflow. This advancement is particularly relevant for engineering applications requiring high geometric accuracy and structural integrity.

Regarding predictive accuracy, the current model achieves an average warpage deviation of 10.6%, which is comparable to the 12% deviation reported by Armillotta et al. [41] and the 9.5% error observed in multi-material simulations by Syrlybayev et al. [5]. Recent studies, such as those by Yu et al. [26], also demonstrate strong agreement between simulation and experiment, particularly when considering infill line directions and their impact on mechanical properties. These comparisons confirm that the model performs within the expected accuracy range for thermo-mechanical FFF simulations.

The thermal modeling strategy (Figure 5) is crucial for capturing the transient heat transfer mechanisms that lead to residual stress formation and warpage. By differentiating between conduction, convection, and radiation effects at each stage of the FFF process—including the critical detachment phase—the model accounts for the evolving thermal boundary conditions that govern the formation of residual stress and warpage. This level of detail, which is often simplified or omitted in earlier studies [16,17], improves the adequacy of predicted temperature fields and, consequently, the mechanical response of the printed part. The explicit simulation of the detachment stage within the staged approach enables the analysis of unsupported regions, where warpage is most pronounced and where failure risks are highest.

Despite these strengths, the model has several limitations. The proposed modeling concept assumes that the thickness of the deposited layer is known and remains constant, as this parameter directly determines the height of the corresponding finite elements in the simulation. While this assumption is reasonable for relatively thin or simple geometries, it may become inadequate as the height and complexity of FFF structures increase. Therefore, further experimental investigations are essential to validate the accuracy of this simplification, particularly for tall or intricate printed components.

Additionally, the current model does not account for viscoelastic or time-dependent material behavior, which can be particularly significant for polymers near their glass transition temperature [44]. Additionally, the current implementation assumes isotropic material properties, and it does not account for interlayer bonding variability or environmental influences, such as ambient temperature and humidity. Future research should aim to address these limitations by incorporating more advanced material models, including temperature-dependent viscoelasticity and anisotropy, to improve the fidelity of simulations and deepen the understanding of deformation mechanisms. The integration of machine learning (ML) techniques also presents a promising direction. ML models trained on simulation and experimental datasets could facilitate rapid prediction of warpage and residual stresses for new geometries and process parameters, significantly reducing computational costs. Furthermore, coupling ML with topology optimization could support the design of structurally optimized and distortion-resistant components.

## 5. Conclusions

This study presents a staged computational framework for simulating the fused filament fabrication (FFF) of polylactic acid (PLA) components, explicitly modeling the sequential printing, cooling, and detachment phases through a coupled thermo-mechanical finite element (FE) approach. From a materials engineering standpoint, the framework addresses key limitations that affect the mechanical performance and dimensional precision of FFF-produced components, including residual thermal stresses, warpage, and interlayer debonding caused by complex and transient thermal gradients during the additive manufacturing (AM) process. The principal contributions of this work are as follows:A staged simulation approach that explicitly incorporates the printing, cooling, and detachment phases, enabling realistic prediction of deformation mechanisms and failure risks throughout the entire fabrication process;An automated procedure for converting G-code into a time-resolved event series, enabling efficient and reproducible simulation workflows through progressive element activation in FE simulations;A transient thermal model coupled with a mechanical simulation to predict residual stresses and warpage;An experimentally validated mechanical model with an average warpage prediction deviation of 10.6%;Sensitivity analyses that confirm the critical influence of simulation parameters—particularly extrusion temperature, mesh density, and time step—on model accuracy;A practical meshing guideline recommending one finite element per filament cross-section to capture thermal and mechanical gradients adequately.

By combining physics-based modeling with automated simulation workflows, the framework provides a scalable and reproducible approach for predicting deformation mechanisms in FFF. The model’s ability to replicate experimentally observed warpage patterns confirms its practical utility for design validation and process optimization. The insights gained from this study can be applied to both prototyping and the manufacturing of functional parts with complex shapes, contributing to the advancement of AM technologies.

Despite certain limitations—such as the assumption of isotropic material behavior and the exclusion of viscoelastic effects—the model establishes a robust foundation for future enhancements. These simulations may include incorporating temperature-dependent anisotropy, realistic interlayer bonding models, and environmental factors. An additional limitation is the assumption that the thickness of the deposited layer is known and remains constant, as this parameter directly determines the height of the corresponding finite elements in the simulation. This simplification may become inadequate for tall or complex FFF structures; therefore, further experimental validation is required to assess its accuracy in such cases. The integration of machine learning techniques could accelerate simulation workflows and enable predictive modeling across a broader range of geometries and process conditions. Future research should aim to overcome these limitations and explore advanced strategies to improve the FFF process, including the use of more realistic material models, optimization of process parameters through machine learning techniques, and investigation of environmental effects on part quality.

## Figures and Tables

**Figure 1 materials-18-04537-f001:**
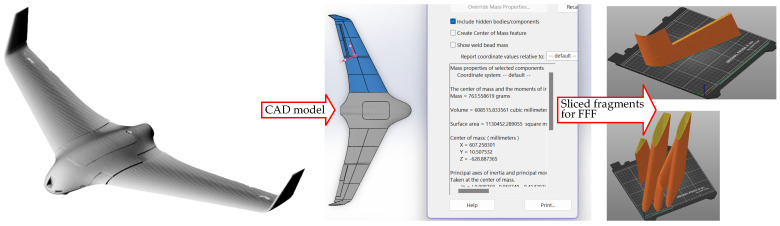
Transforming a drone prototype into sliced wing fragments for FFF (adapted from [27]).

**Figure 2 materials-18-04537-f002:**
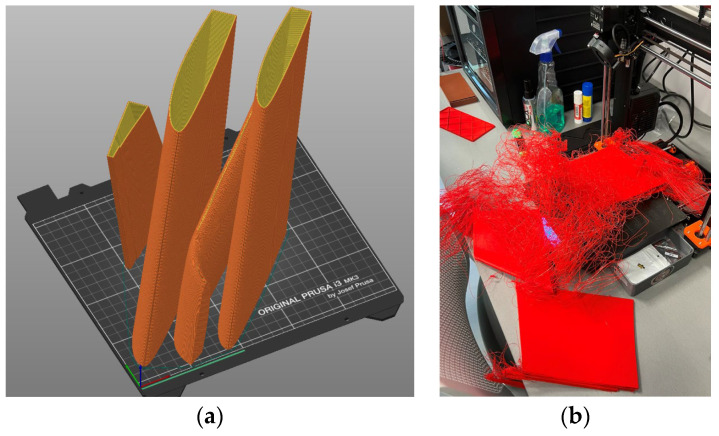
Fabricating wing fragments [27]: (**a**) sliced parts; (**b**) failure of the FFF process.

**Figure 3 materials-18-04537-f003:**
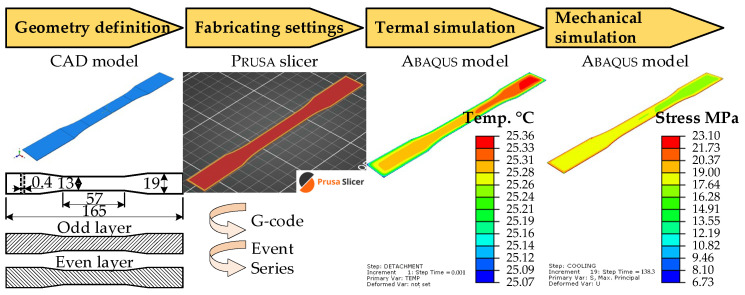
Research workflow.

**Figure 4 materials-18-04537-f004:**
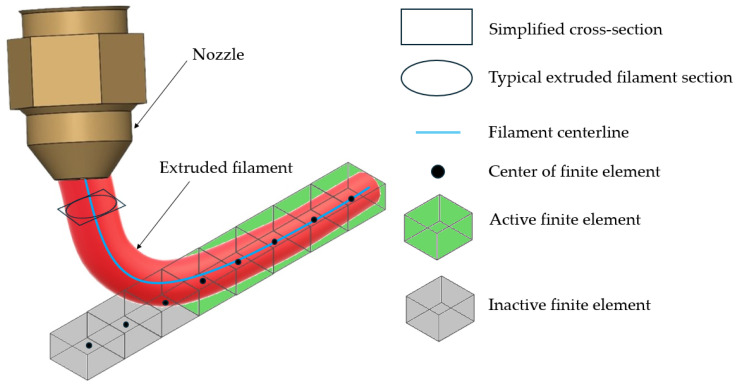
Element activation approach.

**Figure 5 materials-18-04537-f005:**
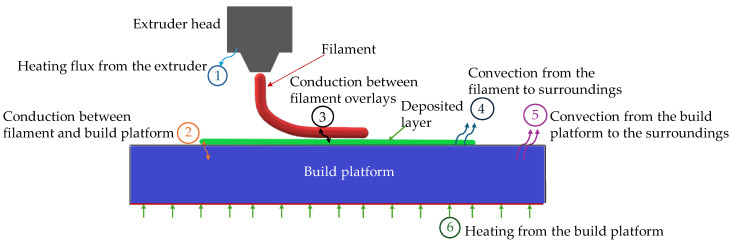
Heat transfer scenarios during the FFF process.

**Figure 6 materials-18-04537-f006:**
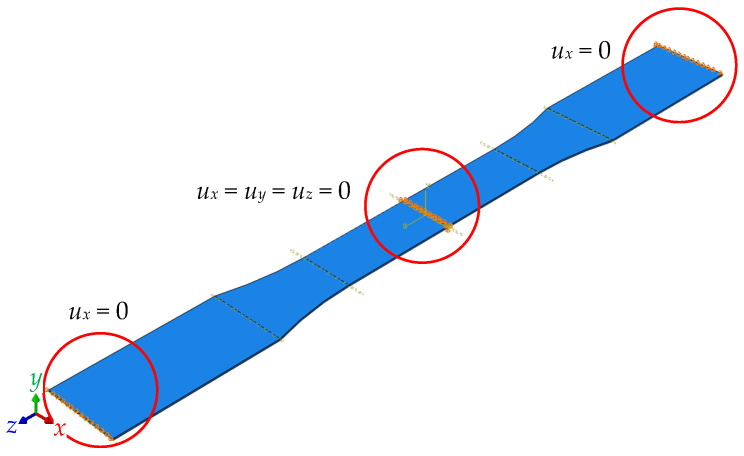
The detachment boundary conditions.

**Figure 7 materials-18-04537-f007:**
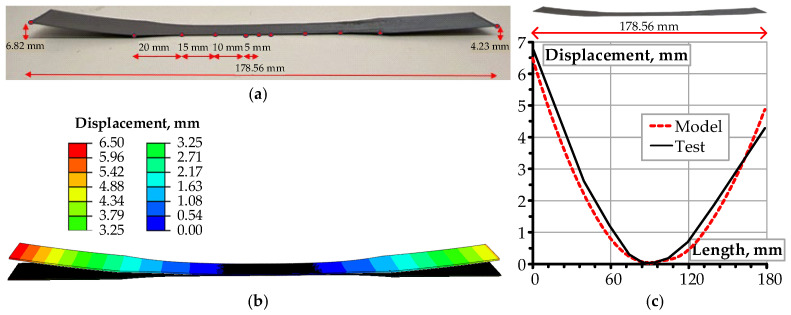
The warping analysis: (**a**) experimental measurement points (red circles); (**b**) FE simulation results; (**c**) the comparative analysis of the deformation profiles.

**Figure 8 materials-18-04537-f008:**
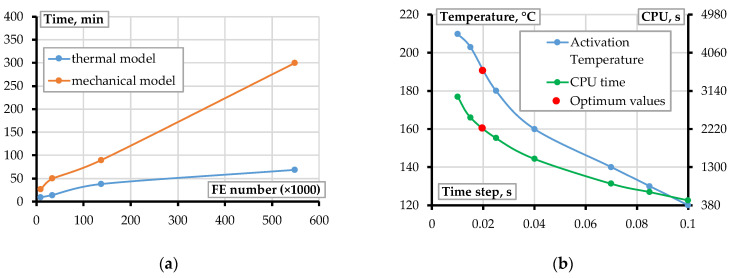
FE mesh optimization: (**a**) the FE number effect on the simulation duration; (**b**) the effect of extrusion temperature and assumed time step on the calculation cost.

**Table 1 materials-18-04537-t001:** Comparative literature analysis related to the thermomechanical simulation of FFF.

Ref.	Software	Scope *	Activation	Detachment	Main Outputs and Validation
[5]	ANSYS	TM	Yes	No	Warpage prediction; validated experimentally
[21]	Abaqus	TM	Yes	Partially	Deformation, residual stress; limited large-scale validation
[22]	ANSYS	TM	Yes	No	Warpage/distortion simulation; validated experimentally
[23]	Digimat	TM	Yes	No (perfect bond)	Warpage/deflection (infill effects); validated experimentally
[24]	Abaqus	M	No (test stage)	No	Tensile behavior across raster angles; no warpage validation
[25]	ANSYS	TM	Yes (birth-death)	No	Warpage prediction; numerical validation
[26]	ANSYS	TM	Yes (birth-death)	No	Warpage/mechanical properties; validated experimentally
This study	Abaqus	TME	Yes	Yes	Warpage modeling (mesh/time-step sensitivity); validated experimentally

* TM = thermomechanical; M = mechanical testing; TME = thermomechanical extended (printing→cooling→detachment).

**Table 2 materials-18-04537-t002:** A fragment of an event series input example.

*t* [s]	*x* [mm]	*y* [mm]	*z* [mm]	Activation
0	–8.75	0.20	49.74	0
0.0003	−8.75	0.20	49.74	1
0.0096	–8.75	0.20	89.25	1
0.0138	8.75	0.20	89.25	1
0.0231	8.75	0.20	49.74	1
0.0234	8.45	0.20	48.68	1
…	…	…	…	…

**Table 3 materials-18-04537-t003:** Physical properties of PLA material.

*T* [°C]	*E* [MPa]	*ν* [–]	*σ_y_* [MPa]	α × 10^–5^ [1/°C]	*k* [W/(m∙°K)]	*c* [J/(kg∙°K)]	*ρ* [kg/m^3^]
25	1860	0.36	25.0	7.9	0.11	1590	1250
30	1800	0.36	25.0	7.9	0.11	1590	1250
40	1727	0.36	25.0	7.9	0.11	1590	1250
50	1603	0.36	18.9	7.9	0.11	1590	1250
60	1000	0.36	15.3	7.9	0.11	1750	1250
70	300	0.36	11.7	7.9	0.11	2300	1250
80	50	0.36	8.2	7.9	0.11	1590	1250
90	10	0.36	7.0	7.9	0.11	1590	1250
100	1 *	0.36	7.0	7.9	0.11	1950	1250
150	1 *	0.36	7.0	7.9	0.11	1950	1250
200	1 *	0.36	7.0	14.4	0	1950	1250
210	1 *	0.36	7.0	14.4	0	1950	1250
220	1 *	0.36	7.0	14.4	0	1950	1250

* Assumed a non-zero value to ensure the solution converges.

## Data Availability

The original contributions presented in this study are included in the article. Further inquiries can be directed to the corresponding author.

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
