# Peer review of "Thermo-Mechanical Approach to Material Extrusion Process During Fused Filament Fabrication of Polymeric Samples"

_materials, 2025, doi:10.3390/ma18194537_

Round 1

Reviewer 1 Report

Comments and Suggestions for Authors

Minor comments

  • Alphabetize the list of keywords.
  • In Table 2, replace all symbols with their full, standardised notations.
  • Provide a detailed justification for the selected extrusion temperature, either supplier recommendations, prior experimental expertise, or systematic optimization studies.
  • Report procurement details of commercial PLA filament.
  • Specify the printing-bed material, as this influences dimensional stability and warpage. Note that PLA typically exhibits minimal warpage with proper printing; identify that the experiments were conducted in an open-chamber setup rather than a closed one.
  • In Figure 5, label the component as “extruder head” rather than simply “extruder.”
  • Figure 7: Report clearly the thickness of the printed sample.
  • Describe the method used to detach the printed sample from the build platform (e.g., cooling-induced release, use of a spatula, or other mechanical means).
  • Report the extrusion and build-plate temperatures clearly in the caption.

Major comments

  • Replace all instances of “fused filament fabrication (FFF)” with “material extrusion (MEX) with filaments” throughout the manuscript to comply with ASTM/ISO 52900:2021 terminology.
  • Line 329: Please justify why you chose the upper and lower temperature limits and explain why you decided to test at 220 °C.
  • Explain why you have chosen this particular shape for printing. Does warpage affect other geometries (e.g. circular or simple shapes) in the same way or is it only present in this design? Have you tested more complicated geometries to assess the warpage behaviour? Typically, increasing print temperature leads to excessive material flow and loss of dimensional accuracy rather than warpage. As warpage is more influenced by material properties and bed temperature, have you evaluated the effects of different bed temperatures?
  • Recommendation for additional testing: I suggest conducting an additional study and corresponding FEM model on the effect of bed temperature on warpage. Including this analysis along with considerations of chamber conditions (open vs. closed, humidity, etc.) would deepen the impact of the manuscript and provide readers with more comprehensive insights.
  • Line 339-341. The photographs suggest that the sample consists of only a few layers. Removal from the build plate can lead to residual stresses that deform the part. Please describe your removal procedure (e.g. the force required) and indicate whether glue or other fixative spray was used to better adhere the first layer.
  • According to ASTM D638‑14, the specimen thickness should be approximately 14 mm. However, your sample, with 11 layers totalling 3.2 mm, does not meet this requirement. Please confirm adherence to the cited standard. Additionally, clarify why you used a 0.2 mm first layer but 0.3 mm for subsequent layers.

Author Response

Acknowledgments: The Authors express their sincere gratitude to the Reviewer for sharing his/her knowledge, thoughtful suggestions, and raised questions, which substantially improved the presentation quality and identified problems for further research. They revised the manuscript, incorporating the Reviewer’s constructive comments; the yellow color highlights the modifications made to the text. In the reply below, the Authors do not distinguish the so-called “minor” and “major” comments, replying to the criticism in a comment-by-comment manner.

1) Comment. Alphabetize the list of keywords.

Answer. The Authors appreciate this comment.

Correction in the manuscript. The Authors reordered the keywords.

2) Comment. In Table 2, replace all symbols with their full, standardised notations.

Answer. This comment is understandable. However, the limited width of the table columns does not allow for explaining these notations inside the table. At the same time, the manuscript includes these explanations near Table 2 (in Lines 254-256). The Authors believe that the current explanations are sufficient and definitive. Therefore, they decided not to react to this comment.

Correction in the manuscript. No corrections were made in this regard.

3) Comment. Provide a detailed justification for the selected extrusion temperature, either supplier recommendations, prior experimental expertise, or systematic optimization studies.

Answer. The Authors sincerely acknowledge this note. According to the manufacturer’s data sheet for the Prusament PLA filament (reference [26] in the updated list of references), a recommended nozzle temperature of 210 ±â€¯10 °C was provided. This defined a practical extrusion temperature range of 200 °C to 220 °C, which was adopted in this study. The previous tests conducted by the Authors’ research team (references [22, 27, 28] in the updated list of references) determine the specific printing settings.

Correction in the manuscript:

  • Lines 163-169. This text describes the PLA fabrication settings and provides a choice for the printing bed temperature.
  • The literature list was extended by adding references [26-28].

4) Comment. Report procurement details of commercial PLA filament.

Answer. This comment aligns with the previous note, and the above answer partially covers the raised issue. Additionally, the Authors would like to mention that the literature analysis [22] substantiated the material choice for this investigation.

Correction in the manuscript. The corresponding clarification was added in Line 163 of the updated manuscript.

5) Comment. Specify the printing-bed material, as this influences dimensional stability and warpage. Note that PLA typically exhibits minimal warpage with proper printing; identify that the experiments were conducted in an open-chamber setup rather than a closed one.

Answer. The Authors appreciate this note. A replacement spring steel sheet with a smooth, double-sided PEI (polyethylenimine) coating was used to fabricate the test sample; no additional measures were taken to enhance adhesion to the printing surface. The physical experiment was conducted in an open-chamber environment.

Correction in the manuscript. The corresponding clarification was introduced in Lines 170-173 of the updated manuscript.

6) Comment. In Figure 5, label the component as “extruder head” rather than simply “extruder.”

Correction in the manuscript. The suggested correction was made.

7) Comment. Figure 7: Report the thickness of the printed sample.

Answer. The Authors acknowledge this essential presentation gap. The experimental trials were conducted in two stages, varying the thickness of the test specimens. The first stage employed a 3.2 mm-thick, dumbbell-shaped test sample, which resulted in barely detectable warpage. Therefore, the second testing trial utilizes the 0.4 mm-thick specimen to stimulate warpage.

Correction in the manuscript:

  • Lines 185-188. This text was rewritten to distinguish the testing stages.
  • Lines 337-338. This reference to Section 3.1 was added to clarify the specimen’s geometry (thickness).

8) Comment. Describe the method used to detach the printed sample from the build platform (e.g., cooling-induced release, use of a spatula, or other mechanical means).

Answer. Initially, the built specimens were cooled in the laboratory conditions (at 20 °C). Once the printing bed temperature reached 25 °C, the sample removal method involved bending (flexing) the smooth PEI-coated spring steel sheet to mechanically release the specimen from the printing surface, where the specimen edge has experienced debonding due to the cooling-induced release.

Correction in the manuscript. The corresponding clarification was introduced in Lines 168-171 of the updated manuscript.

9) Comment. Report the extrusion and build-plate temperatures clearly in the caption.

Answer. The Authors assume that this comment is related to Figure 7. Thus, they consider that this information will be excessive after adding the comment that links the experimental validation to the description of the test specimens in Section 3.1 (see Lines 337-338 of the updated manuscript).

Correction in the manuscript. No additional corrections to those described in replying to Comment 7 above were made in this regard.

10) Comment. Replace all instances of “fused filament fabrication (FFF)” with “material extrusion (MEX) with filaments” throughout the manuscript to comply with ASTM/ISO 52900:2021 terminology.

Answer. The Authors are sincerely grateful for raising this meaningful issue. In this regard, they conducted a representative search in the Google Scholar and Web of Science databases, which delivered the following representative results. The keywords “Additive manufacturing FFF,” “Additive manufacturing FDM,” and “Additive manufacturing MEX” have resulted in approximately 25,200, 72,300, and 24,500 Google Scholar matches. Limiting the search results by publications not older than 2021 has resulted in 15,500, 17,500, and 16,600 matches. In other words, all these terms are almost equally applicable in relatively new publications. However, this search in the Web of Science database, using the above keywords as a research topic, has resulted in 1,801, 3,836, and 450 publications, which informed the final selection of the terminology most commonly used in scientific databases. Thus, the Authors respectfully declined the use of the required “MEX” terminology and replaced “FFF” with “FDM” throughout the manuscript, as “FDM” is the most common term in the scientific literature.

At the same time, the Authors appreciate the suggested reference to the standard terminology. Therefore, they added the corresponding note in Lines 39-43 of the updated manuscript.

Correction in the manuscript:

  • The Authors replaced the “fused filament fabrication (FFF)” terminology with “fused deposition modeling (FDM)” throughout the manuscript.
  • Lines 39-43. The following text was added to introduce the standard terminology: “This technology is also known as fused filament fabrication (FFF) and material extrusion (MEX) with filaments. The international standard ISO/ASTM 52900 [3] recognizes the latter terminology as the standard description of this fabrication process. Nevertheless, this manuscript adopts the FDM abbreviation as the most common term in the scientific literature.”
  • The international standard ISO/ASTM 52900 [3] was added to the list of references.

11) Comment. Line 329: Please justify why you chose the upper and lower temperature limits and explain why you decided to test at 220 °C.

Answer. This note aligns with Comment 3, and the Authors partially clarified the issue by replying to Comment 3 above. The tests were conducted employing the 210 °C nozzle temperature. Still, the Authors acknowledge that the manufacturer’s datasheet for Prusament PLA filament [26] specifies a recommended nozzle temperature range of 210 ±â€¯10 °C, without detailing specific conditions for selecting a particular value within this interval. In this study, the extrusion temperature of 220 °C was chosen to explore the upper bound of the recommended range as part of a sensitivity analysis. This decision was motivated by the need to assess the influence of thermal input on warpage behavior, particularly under open-chamber conditions and reduced specimen thickness. The selected temperature remains within the manufacturer’s allowable range and reflects a practical scenario encountered in desktop FDM applications.

Correction in the manuscript:

  • Lines 163-169. This text describes the PLA fabrication settings and provides a choice for the printing bed temperature.
  • Line 353-355. The following comment was added to clarify the issue: “…as recommended by the manufacturer’s datasheet [26]. Within the manufacturer’s allowable range, the selected temperatures reflect a practical scenario encountered in desktop FDM applications.”

12 & 13) Comments. Explain why you have chosen this particular shape for printing. Does warpage affect other geometries (e.g., circular or simple shapes) in the same way, or is it only present in this design? Have you tested more complicated geometries to assess the warpage behavior? Typically, increasing the print temperature leads to excessive material flow and loss of dimensional accuracy, rather than warping. As material properties and bed temperature have a greater influence on warpage, have you evaluated the effects of different bed temperatures? Recommendation for additional testing: I suggest conducting an additional study and corresponding FEM model on the effect of bed temperature on warpage. Including this analysis along with considerations of chamber conditions (open vs. closed, humidity, etc.) would deepen the impact of the manuscript and provide readers with more comprehensive insights.

Answer. The Authors sincerely appreciate these insightful comments and suggestions. They encountered warpage problems during drone prototyping, as described in Section 2. However, both modeling and experimental verification of the prediction results are challenging due to the complexity of such geometries, as shown in Figure 2. Therefore, this study employs the ASTM D638 Type I tensile specimen geometry as a standardized and repeatable shape that offers both mechanical relevance and sensitivity to thermal deformation, such as warpage. The dumbbell shape creates a high aspect ratio and long unsupported span, making it particularly prone to edge lifting and curling during the FDM process.

At the same time, the Authors agree that warpage can vary significantly across different geometries. Still, this study focuses on specimen geometry, a widely used concept in mechanical testing that has practical relevance for designing functional parts. In this initial phase, the Authors had not tested more complex or curved geometries (e.g., circular, lattice, or overhang structures). However, the approach and findings presented here can be extended to such geometries in future research.

Regarding temperature parameters, the Authors also agree that print temperature primarily affects material flow, interlayer bonding, and dimensional accuracy, rather than being the dominant factor in warpage. In this study, warpage was found to be more significantly influenced by the transient thermal gradients, low part thickness, and cooling in an open chamber. Although the printing bed temperature was set within the manufacturer’s recommended range (40–60 °C), the Authors recognize that varying the bed temperature could influence the adhesion dynamics and residual stress release. However, the effects of different bed temperatures were not systematically studied in this work and are considered, together with the offered tests, as outlines for future investigation.

Correction in the manuscript. No corrections were made in this regard.

14) Comment. Line 339-341. The photographs suggest that the sample consists of only a few layers. Removal from the build plate can lead to residual stresses that deform the part. Please describe your removal procedure (e.g., the force required) and indicate whether glue or other fixative spray was used to adhere the first layer better.

Answer. This query aligns with Comments 5 and 8, and the Authors clarified this issue by answering those questions.

Correction in the manuscript. The new text on Lines 185-188 clarifies the manufacturing conditions.

15) Comment. According to ASTM D638-14, the specimen thickness should be approximately 14 mm. However, your sample, with 11 layers totalling 3.2 mm, does not meet this requirement. Please confirm adherence to the cited standard. Additionally, clarify why you used a 0.2 mm first layer but 0.3 mm for subsequent layers.

Answer. The Authors respectfully disagree with the eminent Reviewer, since the comment to the dimension table on Page 4 of this standard, among others, states that “Thickness, T, shall be 3.2 ± 0.4 mm (0.13 ± 0.02 in.) for all types of molded specimens, and for other Types I and II specimens where possible.”

This choice of layer thickness configuration was intentional and based on a common FDM printing practice that improves adhesion, stability, and print efficiency. A thickness of 0.2 mm for the first layer was selected to ensure strong and uniform adhesion to the build platform. A slightly thinner first layer enhances the contact area between the extruded filament and the bed surface, which is essential for long and narrow parts, such as the D638 specimen. For the remaining layers, a thickness of 0.3 mm was used to increase printing speed and reduce total build time, while maintaining sufficient mechanical integrity. Since the first layer has been considered a stable foundation, the printer can deposit thicker layers without compromising layer bonding or dimensional accuracy for the rest of the part. At the same time, the simplified specimen used for warpage analysis consisted of two layers of equal 0.2 mm thickness, as described in Section 3.1 of the updated manuscript.

Correction in the manuscript. The following sentence was added in Lines 185-188 of the updated manuscript: “To observe warpage at the macroscale, the second fabrication stage utilized the same fabrication parameters as the first stage, but with a reduced thickness of 0.4 mm, printed in two layers of 0.2 mm each.”

Reviewer 2 Report

Comments and Suggestions for Authors

The manuscript entitled “Thermo-Mechanical Approach to Material Extrusion Process during Fused Filament Fabrication of Polymeric Samples” constructed a thermos-mechanical coupled finite element framework based on G-code automatic conversion, which was used to predict the warpage and residual stress of PLA during FFF process. With experimental verification error at about 10.6%, this model has good engineering applicability. This study directly mapped the printing path to unit activation events, effectively improving simulation efficiency and scalability. This paper is recommended for publication in Materials after addressing following issues:     

  1. The mechanical mesh model accuracy verification results in Figure 8a showed an increasing trend, which seems to be not converging. The author should give detailed explanation about this phenomenon.
  2. The model did not consider the heat conduction between the support and the component, which may influent the result of the warped displacement.
  3. The English expression needs optimization.

Author Response

Acknowledgments: The Authors express their sincere gratitude to the Reviewer for sharing his/her time and knowledge; thoughtful suggestions, which substantially improved the presentation quality, are genuinely appreciated.

1) Comment. The manuscript entitled “Thermo-Mechanical Approach to Material Extrusion Process during Fused Filament Fabrication of Polymeric Samples” constructed a thermo-mechanical coupled finite element framework based on G-code automatic conversion, which was used to predict the warpage and residual stress of PLA during the FFF process. With an experimental verification error of approximately 10.6%, this model exhibits good engineering applicability. This study directly mapped the printing path to unit activation events, effectively improving simulation efficiency and scalability. This paper is recommended for publication in Materials after addressing the following issues…

Answer. The Authors sincerely appreciate the favorable evaluation of this study. They revised the manuscript, incorporating the Reviewer’s constructive comments.

Correction in the manuscript. The yellow color highlights the modifications made to the text.

2) Comment. The mechanical mesh model accuracy verification results in Figure 8a show an increasing trend, which does not seem to be converging. The author should give a detailed explanation about this phenomenon.

Answer. The Authors appreciate this thoughtful observation. Developing an appropriate meshing strategy is not straightforward and must be tailored to the specific needs of the analysis. While finer mesh enhances the accuracy of stress gradient predictions, it also leads to a substantial increase in computational cost. This contradiction is evident in mechanical analyses, where stress concentration effects make the system more sensitive to mesh refinement. In this model, the observed increase in accuracy does not appear to converge as the number of elements increases. This phenomenon often arises from local stress singularities or, where stress continues to rise with mesh refinement, giving the impression of non-convergence. Additionally, refining the mesh uniformly without focusing on critical stress regions can be inefficient and misleading, even though it may still result in increased computational time. In some cases, nonlinear material behavior can also affect this trend.

Therefore, for smaller models, where local stress effects are significant and computational costs remain manageable, a finer mesh is recommended to capture detailed mechanical responses adequately. In contrast, for larger models where local variations are minor compared to the overall geometry, a coarser mesh is generally more suitable for reducing computational demands while still maintaining acceptable accuracy. A balanced meshing approach, possibly involving local refinement and adaptive strategies, is essential to ensure both computational efficiency and reliable simulation results.

Correction in the manuscript:

  • The corresponding discussion was added in Lines 419-434 of the updated manuscript.
  • References [38] and [39] were added to substantiate this discussion.

3) Comment. The model did not account for heat conduction between the support and the component, which may affect the result of the warped displacement.

Answer. The Authors appreciate this insightful comment. The presence of the print bed temperature was simulated as a temperature boundary condition to the bottom surface of the 3D printed object. During the first heat transfer step analysis, the “Deposition phase,” a constant 60 °C boundary condition is applied to the bottom surface. In contrast, this boundary condition is inactive during the cooling and detachment step analysis. The simulation of the typical contact between the print bed/build platform and the 3D-printed component is challenging and requires complex experimental measurements. It describes the future investigation object.

Correction in the manuscript. No corrections were made in this regard.

4) Comment. The English expression needs optimization.

Answer. The Authors accept this criticism and have entirely verified the manuscript, improving the writing style and clarity to the best of their knowledge.

Correction in the manuscript. The Authors have entirely verified the manuscript. The yellow color highlights all modifications in the text.

Reviewer 3 Report

Comments and Suggestions for Authors

This paper presents a study on a thermo-mechanical approach for the numerical simulation of fused deposition modeling (FDM) of PLA samples. The research combines numerical and experimental analysis, resulting in average deviations of ~10%.

The paper looks promising and presents valuable insights into thermo-mechanical modeling of FDM, but there are still some flaws. However, I believe that these can be reviewed without a complete rework or rewriting of the paper. Some remarks follow:

  1. I believe that the paper is well illustrated, and the Introduction is well written with a very good state-of-the-art review.
  2. Pg.3 ln.102–110: I believe, however, that in Section 2 the sentences in the first paragraph are very short, almost feeling like one is reading a bullet-point list.
  3. Pg.4–7: Sections 3.1 and 3.2 have many repetitions and the text should be rewritten to avoid them. Some examples follow in comments 4–8.
  4. Pg.5 ln.157–159: This sentence feels like a repetition of what was stated in Section 2, so it may not be necessary, or it can be much briefer, so it connects to the next sentence.
  5. Pg.5 ln.198–201: Although it is a bit more complete than before, some of what is explained in these two sentences was already explained in lines 192–195. Maybe the first and second paragraphs of Section 3.2 can be combined so repetitions are avoided.
  6. Pg.6 ln.218–219: Again, this sentence is already stated before, so it shouldn't be here.
  7. Pg.6 ln.220: This ‘derived from the event series’ is better explained in the next sentence, so it is also an unnecessary repetition.
  8. Pg.6 ln.222–225: This was also explained in the previous paragraph.
  9. Fig.3: There is a typo in Figure 3, reading 'PURSA' instead of 'PRUSA'.
  10. Section 3.5: I do not understand why this section is not part of Section 4 instead of 3.

Author Response

Comment 1. The paper looks promising and presents valuable insights into thermo-mechanical modeling of FDM, but there are still some flaws. However, I believe that these can be reviewed without a complete rework or rewriting of the paper. I think the paper is well-illustrated, and the Introduction is well-written, featuring a very good state-of-the-art review.

Reply. The constructive evaluation and positive assessment of the manuscript are sincerely appreciated. The Reviewer’s recognition of the study’s scientific value and the quality of the state-of-the-art review is acknowledged. The comments provided are considered highly constructive and have been carefully addressed further to improve the clarity and presentation of the manuscript. The yellow color highlights all corrections in the text.

Comment 2. Pg. 3 ln.102–110: I believe, however, that in Section 2, the sentences in the first paragraph are very short, almost feeling like one is reading a bullet-point list.

Reply. The Reviewer’s observation regarding the presentation gap in Section 2 is acknowledged. The formulation of the research problem and the description of the innovative aspects of the proposed model have been clarified and expanded to improve the manuscript’s clarity and coherence.

Correction in the manuscript. The problem formulation has been strengthened by presenting it in the first paragraph of Section 2, emphasizing the practical challenges encountered during fabrication. The innovative aspects of the model, particularly the explicit simulation of the detachment stage, are now clearly described. Additionally, the third paragraph of Section 2 has been revised to highlight the advancement over previous approaches and to specify the direct integration of slicing data into element activation.

Comment 3. Pg. 4–7: Sections 3.1 and 3.2 have many repetitions, and the text should be rewritten to avoid them. Some examples follow in comments 4–8.

Reply. The Reviewer’s careful reading of the manuscript and the identification of repetitive content in Sections 3.1 and 3.2 are appreciated. The valuable suggestions for improving the clarity and conciseness of these sections have been carefully considered and addressed in the revised manuscript. Redundant statements were removed, and the text was streamlined as detailed in the following replies.

Comment 4. Pg. 5 ln. 157–159: This sentence feels like a repetition of what was stated in Section 2, so it may not be necessary, or it can be much briefer, so it connects to the next sentence.

Correction in the manuscript. The criticized statements in Section 3.1 have been rephrased to eliminate repetition and to refer explicitly to the problem formulation and innovation already presented in Section 2.

Comment 5. Pg. 5 ln. 198–201: Although it is a bit more complete than before, some of what is explained in these two sentences was already described in lines 192–195. Perhaps the first and second paragraphs of Section 3.2 can be combined to avoid repetition.

Correction in the manuscript. As recommended, the first and second paragraphs of Section 3.2 have been merged and rephrased to eliminate repetition and improve the clarity of the description of the G-code processing and event series generation.

Comment 6. Pg. 6 ln. 218–219: Again, this sentence is already stated before, so it shouldn't be here.

Reply. The Reviewer’s observation regarding repetition is acknowledged. While it is necessary to mention the transformation of G-code into an event series to maintain the logical flow of the methodology, the statement has been revised to refer back to the earlier explanation and avoid unnecessary duplication.

Correction in the manuscript. The criticized sentence has been rephrased as: “As previously described, the custom PYTHON script transforms this G-code into an event series for FE analysis,” thereby ensuring clarity without repeating technical details.

Comment 7. Pg. 6 ln. 220: This ‘derived from the event series’ is better explained in the following sentence, so it is also an unnecessary repetition.

Correction in the manuscript. The criticized sentences were merged to eliminate duplication, and the redundant statement regarding the derivation of coordinates from the event series was deleted for clarity.

Comment 8. Pg. 6 ln. 222–225: This was also explained in the previous paragraph.

Correction in the manuscript. The criticized sentences were revised to remove unnecessary repetition, and the redundant explanation was deleted to improve the clarity and conciseness of the text.

Comment 9. Fig. 3: There is a typo in Figure 3, reading 'PURSA' instead of 'PRUSA'.

Correction in the manuscript. The Reviewer’s careful attention to detail is appreciated. The typographical error in Figure 3 has been corrected as suggested.

Comment 10. Section 3.5: I do not understand why this section is not part of Section 4 instead of 3.

Reply. The Reviewer’s concern regarding the placement of Section 3.5 is understood. While this section could nominally appear as a stand-alone part of the manuscript, a more substantial verification would be necessary to ensure the completeness of the discussion. In the present context, Section 3.5 is intended to illustrate the essential capability of the proposed model—specifically, its ability to simulate the detachment process.

Correction in the manuscript. To clarify the above point, a paragraph has been introduced at the beginning of Section 3.5 to emphasize the significance of these simulations within the overall modeling framework.

Acknowledgements. The Authors sincerely appreciate the Reviewer’s careful reading of the manuscript and the valuable, constructive comments provided. The thoughtful feedback has contributed significantly to improving the clarity, coherence, and overall quality of the work.

Reviewer 4 Report

Comments and Suggestions for Authors

The presented work presents a thermomechanical analysis of PLA shapes created during 3D printing. The topic itself is interesting, but the available literature contains numerous scientific articles detailing the properties of materials that can be used as 3D printing filaments and finished products. Due to the lack of novelty in the reviewed work, I am forced to reject it.

Author Response

Comment 1. The presented work presents a thermomechanical analysis of PLA shapes created during 3D printing. The topic itself is interesting, but the available literature contains numerous scientific articles detailing the properties of materials that can be used as 3D printing filaments and finished products. Due to the lack of novelty in the reviewed work, I am forced to reject it.

Reply. The Reviewer’s concern regarding the novelty of the present work is acknowledged. Although numerous studies have addressed the thermomechanical analysis of 3D-printed materials and components, the essential innovation of this study lies in the development and experimental validation of a staged simulation framework that explicitly models the sequential printing, cooling, and detachment phases in fused filament fabrication (FFF). As summarized in the new Table 1 of the revised manuscript, most existing finite element modeling approaches are limited to the printing and cooling stages and assume perfect adhesion to the build platform throughout the process. The explicit simulation of the detachment stage, where the printed part is released from the build platform and residual stresses are relaxed, remains largely unaddressed in the literature. This staged modeling concept enables a more realistic prediction of warpage and deformation mechanisms, as confirmed by experimental validation on PLA specimens. The proposed approach thus addresses a critical gap in the field and provides new insights for the predictive modeling and optimization of extrusion-based additive manufacturing processes.

In addition, the developed framework is particularly valuable for polymeric materials that exhibit significant shrinkage deformation during the cooling stage. By explicitly simulating the detachment phase, the model can predict unfavorable outcomes—such as excessive warpage or loss of dimensional accuracy—before physical fabrication is attempted. This predictive capability allows for the optimization of process parameters and part geometry in a virtual environment, reducing material waste and experimental effort. The approach therefore offers substantial potential for advancing materials engineering and computer-aided fabrication technologies, supporting the design and manufacture of high-precision, distortion-resistant components in a wide range of polymeric systems.

The Authors acknowledge that the above issues were unclear in the original version of the manuscript. Therefore, they reworked the presentation to highlight the essential contribution of this study to material engineering in general and the numerical simulation of FFF in particular. The text was substantially modified, and the yellow color highlights all corrections in the manuscript.

Corrections in the manuscript: To clarify the novelty and fundamental contribution of this study, the manuscript was substantially revised, and all modifications are highlighted in yellow. The Introduction was restructured to define the research gap explicitly and to emphasize the unique aspect of the staged simulation framework, which incorporates the sequential modeling of printing, cooling, and, critically, the detachment phase. A new comparative Table 1 was introduced to systematically benchmark the present approach against recent literature, clearly demonstrating that explicit simulation of the detachment stage remains unaddressed mainly in prior studies. The motivation for the research (Section 2) was expanded to highlight the practical necessity of detachment modeling for accurate prediction of warpage and residual stress release in FFF-fabricated components. The discussion in Section 4.1 was extended to provide a detailed explanation of the staged simulation concept, with particular attention to the experimental validation of the detachment phase and its impact on predictive accuracy. The Conclusions were revised to accentuate the essential innovation of the research and to articulate its implications for advancing materials engineering and computer-aided fabrication technologies. These comprehensive modifications were implemented to ensure that the manuscript now clearly demonstrates the original scientific contribution and addresses the critical gap identified by the Reviewer.

Comment 2. Does the introduction provide sufficient background and include all relevant references? Must be improved

Correction in the manuscript. The Introduction was revised to expand the background discussion and ensure inclusion of all relevant references, thereby strengthening the contextual foundation for the study.

Comment 3. Is the research design appropriate? Must be improved

Reply. The definition of the research innovation was clarified in the revised manuscript to ensure that the scope, objectives, and methodological approach are explicitly stated and aligned with the study’s aims.

Correction in the manuscript. The relevant sections (the Introduction, Sections 2, 4.1, and 4.3, and Conclusions) were updated to sharpen the definition of the research innovation and to clarify the methodological framework, ensuring that the research design is clearly articulated.

Comment 4. Are the conclusions supported by the results? Can be improved

Correction in the manuscript. The Conclusions section was revised to ensure that all statements are directly supported by the presented results and to reflect the main findings and their implications clearly.

Round 2

Reviewer 1 Report

Comments and Suggestions for Authors

The authors have provided sufficient justification and have modified the manuscript based on the comments. I suggest that the manuscript be accepted for publication.

Author Response

Comment. The authors have provided sufficient justification and have modified the manuscript based on the comments. I suggest that the manuscript be accepted for publication.

Reply. The careful revision and constructive comments provided by the Reviewer are sincerely appreciated. These suggestions have substantially improved the clarity and overall presentation quality of the manuscript.

Reviewer 4 Report

Comments and Suggestions for Authors

The text has been significantly revised, and the feedback has been incorporated. The manuscript now merits acceptance.